# Dynamic Parameter Optimization for Highly Transferable Transformation-based Attacks

## Abstract

Despite their wide application, the vulnerabilities of deep neural networks raise societal concerns. Among them, transformation-based attacks have demonstrated notable success in transfer attacks. However, existing attacks suffer from blind spots in parameter optimization, limiting their full potential. Specifically, (1) prior work generally considers low-iteration settings, yet attacks perform quite differently at higher iterations, so characterizing overall performance based only on low-iteration results is misleading. (2) Existing attacks use uniform parameters for different surrogate models, iterations, and tasks, which greatly impairs transferability. (3) Traditional transformation parameter optimization relies on grid search. For $n$ parameters with $m$ steps each, the complexity is $\mathcal{O}(mn)$. Large computational overhead limits further optimization of parameters. To address these limitations, we conduct an empirical study with various transformations as baselines, revealing three dynamic patterns of transferability with respect to parameter strength. We further propose a novel Concentric Decay Model (CDM) to effectively explain these patterns. Building on these insights, we propose an efficient Dynamic Parameter Optimization (DPO) based on the rise-then-fall pattern, reducing the complexity to $\mathcal{O}(n \log_2 m)$. Comprehensive experiments on existing transformation-based attacks across different surrogate models, iterations, and tasks demonstrate that our DPO can significantly improve transferability. The code will be released to the public.

## 1 Introduction

Deep neural networks (DNNs) He et al. (2016); Dosovitskiy et al. (2020) are being increasingly applied Xia et al. (2025); Arhouni et al. (2025), yet adversarial examples Szegedy et al. (2013); Goodfellow et al. (2014) formed by superimposing imperceptible perturbations to benign samples can fool them with high confidence, posing great real-world threats Thummala et al. (2024); Li et al. (2024). To support reliable deployment, proactively identifying potential attacks is necessary.

In practice, black-box attacks, where the adversary knows neither the target model's structure nor its parameters, represent substantial threats. Among these, transformation-based attacks Xie et al. (2019); Wang et al. (2021) are a promising branch. They improve transferability Papernot et al. (2016) by applying multiple input transformations, backpropagating to obtain several perturbation copies, and averaging them to stabilize the perturbation. Prior studies Lin et al. (2024); Wang et al. (2024a) have made considerable progress. Unfortunately, they focus on the effects of transformation types on transferability while neglecting the dynamics on parameters, which creates blind spots in parameter optimization and leads to its insufficiency, thereby undermining transferability. To bridge this gap, we endeavor to explore the dynamics of transferability with respect to parameters.

To derive generalizable insights, we commence with a quantitative analysis of representative transformations widely adopted in state-of-the-art attacks, including *Translation* Wang et al. (2023); Zhu et al. (2024), *Block Shuffle* Wang et al. (2024a); Zhu et al. (2024), *Rotation* Wang et al. (2023; 2024a); Zhu et al. (2024); Guo et al. (2025), *Noise Addition* Wang et al. (2023); Zhu et al. (2024), and *Resize* Zhu et al. (2024); Guo et al. (2025). Building on these transformations, we discretely vary parameters to generate adversarial examples under different surrogates and iterations, subsequently evaluating their transferability. Multiple runs with different seeds show us three patterns:

(i) Transformations leading at low iterations may not retain their advantage at higher iterations.

(ii) Transferability growth over iterations and surrogates differs markedly across parameters. Generally, optimal transformation magnitudes grow with iterations and vary across surrogates.

(iii) Transferability follows a rise-then-fall pattern with respect to transformation magnitude.

These patterns provide empirical support for the Model Augmentation Theory Wang et al. (2021), a well-established explanation for transformation-based attacks. Simultaneously, these observations illuminate the landscape surrounding the surrogate model, inspiring us to propose *Concentric Decay Model* (CDM). Specifically, the Model Augmentation Theory posits that composing transformations with a surrogate model emulates diverse models, generating perturbations that are less prone to overfitting the surrogate and thus transfer better. Our Concentric Decay Model further proposes that plausible models (i.e., those more consistent with real models (see Section 3.3)) around the surrogate model are concentrically distributed with density decreasing outward in a space defined by KL divergence. Within this framework, the above three patterns are easily explained (see Section 3.3).

Furthermore, our Concentric Decay Model and empirical observations suggest that transformation parameters should be dynamically evolved. That is, the optimal parameters of transformation-based attacks vary with the surrogate, iteration, and task. However, existing attacks generally configure identical parameters, a practice that violates this principle and impairs transferability. Besides, previous studies typically compare transferability under insufficient low-iteration settings, such as 10 Dong et al. (2019); Wang et al. (2021); Long et al. (2022); Wang et al. (2023); Zhang et al. (2023); Wang et al. (2024a); Lin et al. (2024); Guo et al. (2025), 16 Lin et al. (2019), or 30 Xie et al. (2019) epochs. Low-iteration performance cannot reliably reflect the landscape at high iterations, which underestimates the overall effectiveness of the attack. Therefore, we propose an efficient *Dynamic Parameter Optimization* (DPO) and validate it across diverse existing transformation-based attacks, including *Admix* Wang et al. (2021), *SSIM* Long et al. (2022), *STM* Ge et al. (2023), and *BSR* Wang et al. (2024a). Comprehensive experiments show that the optimized parameters differ substantially from the official versions and significantly improve transferability. As a representative case with the state-of-the-art *BSR*, with R50 as the surrogate model, the average non-targeted attack success rate (ASR) improves by $3.2\%$ to $90.7\%$ at Epoch 100 across eight diverse test models, while the targeted ASR increases by $13.9\%$ to $34.1\%$ at Epoch 200, with remarkable gains of $34.1\%$ on VGG-16 and $27.1\%$ on RegNet. Moreover, based on the rise-then-fall pattern, we propose a bisection-based DPO. For $n$ parameters with $m$ steps each, it reduces the optimization complexity from $\mathcal{O}(mn)$ to $\mathcal{O}(n \log_2 m)$. This can greatly accelerate parameter optimization for existing or future methods.

The main contributions of this paper are summarized as follows:

- To the best of our knowledge, this is the first work to study the dynamics of attack transferability with respect to transformation parameters, and reveal three dynamic patterns that hold across different transformations.

- We propose a novel Concentric Decay Model (CDM), which describes the distribution of plausible models around the surrogate and effectively explains observations that optimal parameters vary dynamically.

- We propose an efficient Dynamic Parameter Optimization (DPO), which is based on the rise-then-fall pattern and a bisection approach. Our DPO reduces the computational complexity from $\mathcal{O}(mn)$ to $\mathcal{O}(m \log_2 m)$.

- Extensive experiments across various attacks demonstrate that the re-optimized parameters by our DPO yield significant transferability gains.

## 2  RELATED WORK

### 2.1  ADVERSARIAL ATTACKS

Adversarial attacks involve adding imperceptible perturbations to benign samples, subject to certain constraints (e.g., $L_2$- Carlini & Wagner (2017) or $L_\infty$-norm Goodfellow et al. (2014) budgets, or limitations on the number of modified pixels Narodytska & Kasiviswanathan (2016)), such that the resulting adversarial examples can fool classifiers with high confidence. Depending on whether the adversary has complete access to the target model's structure and parameters, adversarial attacks can be classified into white-box and black-box attacks (the intermediate case, often termed gray-box attacks). Iterative gradient-based attacks, such as I-FGSM Goodfellow et al. (2014) and MI-

FGSM Dong et al. (2018), are sufficient to achieve a successful white-box attack. However, the target model is typically black-box in practice. Prior research has investigated multiple strategies to carry out black-box attacks. Query-based strategies Chen et al. (2017); Wang et al. (2025); Reza et al. (2025) iteratively refine the perturbations by leveraging the outputs of the target model over multiple queries. Transfer-based methods Tang et al. (2024); Wang et al. (2024a;b); Guo et al. (2025); Ren et al. (2025) leverage the cross-model effectiveness of adversarial examples to perform attacks. Transformation-based attacks represent a promising branch of transfer-based strategies.

## 2.2 TRANSFORMATION-BASED ATTACKS

Transformation-based attacks apply multiple input transformations per iteration to generate perturbation copies, which are averaged to stabilize the perturbation direction and prevent overfitting the surrogate model. Previous studies devote substantial effort to investigating the impact of transformation types on transferability. DIM Xie et al. (2019), TIM Dong et al. (2019), SIM Lin et al. (2019), and Admix Wang et al. (2021) examine the effects of resizing, translation, scaling, and mixup on transferability, respectively. SSIM Long et al. (2022) investigates the impact of frequency-domain noise on transferability. STM Ge et al. (2023) enhances transferability by leveraging image stylization. SIA Wang et al. (2023) enhances transferability by combining multiple transformations, including translation, flipping, rotation, scaling, noise addition, and blurring. DeCoWA Lin et al. (2024) improves transferability by distorting images. BSR Wang et al. (2024a) improves transferability by applying block shuffle and rotation. L2T Zhu et al. (2024) dynamically selects transformations and leverages a combination of rotation, scaling, resizing, block shuffling, mixup, masking, frequency-domain noise, cropping, and shearing to enhance transferability. However, beyond transformation types, transformation parameters also critically affect transferability. Prior work overlooks the dynamics of transferability with respect to parameters, generally focusing on low-iteration comparisons and configuring identical parameters across different scenarios. To fill this gap, we delve into this intriguing topic, delivering insights from both theoretical and methodological viewpoints.

## 3 METHODOLOGY

### 3.1 PRELIMINARIES

Given an input-label pair $(\boldsymbol{x}, y) \in (\boldsymbol{\mathcal{X}}, \mathcal{Y})$, a surrogate model $f_S \in \mathcal{F}$, and a target model $f_T \in \mathcal{F}$, let $B_p(\boldsymbol{x}, \epsilon) = \{\boldsymbol{x}' \in \boldsymbol{\mathcal{X}} \mid ||\boldsymbol{x}' - \boldsymbol{x}||_p \leq \epsilon\}$ denote the $p$-ball around $\boldsymbol{x}$ with radius $\epsilon$. The attack $A \in \mathcal{A}$ aims to find an adversarial example $\boldsymbol{x}_{adv} = A(\boldsymbol{x}; f_S) \in B_p(\boldsymbol{x}, \epsilon)$ such that

$$\arg\max f_T(\boldsymbol{x}_{adv}) \neq y. \tag{1}$$

Here, $\boldsymbol{\mathcal{X}}$, $\mathcal{Y}$, $\mathcal{F}$, and $\mathcal{A}$ represent the sample, label, model, and algorithm spaces, respectively. Following prior work, we focus on the case $p = \infty$.

Transformation-based strategies augment the surrogate model $f_S$ with a transformation module $\mathcal{T}$ parameterized by a random variable $\boldsymbol{\theta} \sim \boldsymbol{\Theta}(\boldsymbol{z})$, where the transformation parameter $\boldsymbol{z}$ governs the distribution $\boldsymbol{\Theta}$. Thus, transformation-based strategies can be summarized as

$$\boldsymbol{x}_{adv} = A(\boldsymbol{x}; \mathcal{T}(\boldsymbol{z}) \circ f_S) \in B_p(\boldsymbol{x}, \epsilon), \tag{2}$$

where $\circ$ denotes module cascading. Consistent with prior work, $A$ adopts MI-FGSM, and the transformation-based attack can be detailed as:

$$g^{(t)} = \lambda \cdot g^{(t-1)} + \frac{\bar{\mu}}{||\bar{\mu}||_1}, \quad \bar{\mu} = \frac{1}{N} \Sigma_{i=1}^{N} \nabla_{\boldsymbol{x}_{adv}^{(t-1)}} J(f_S(\mathcal{T}(\boldsymbol{x}_{adv}^{(t-1)}; \boldsymbol{z})), y), \tag{3}$$

$$\boldsymbol{x}_{adv}^{(t)} = \boldsymbol{x}_{adv}^{(t-1)} + \alpha \cdot \text{sign}(g^{(t)}). \tag{4}$$

Here, $J$ is the loss function. $N$ is the number of gradient copies. $\boldsymbol{x}_{adv}^{(t)}$ is the $t$-iteration ($0 \leq t \leq T$) example. $\lambda$ is the momentum factor. $\alpha = \epsilon/T$ is the step size. $\text{sign}(\cdot)$ denotes the sign function.

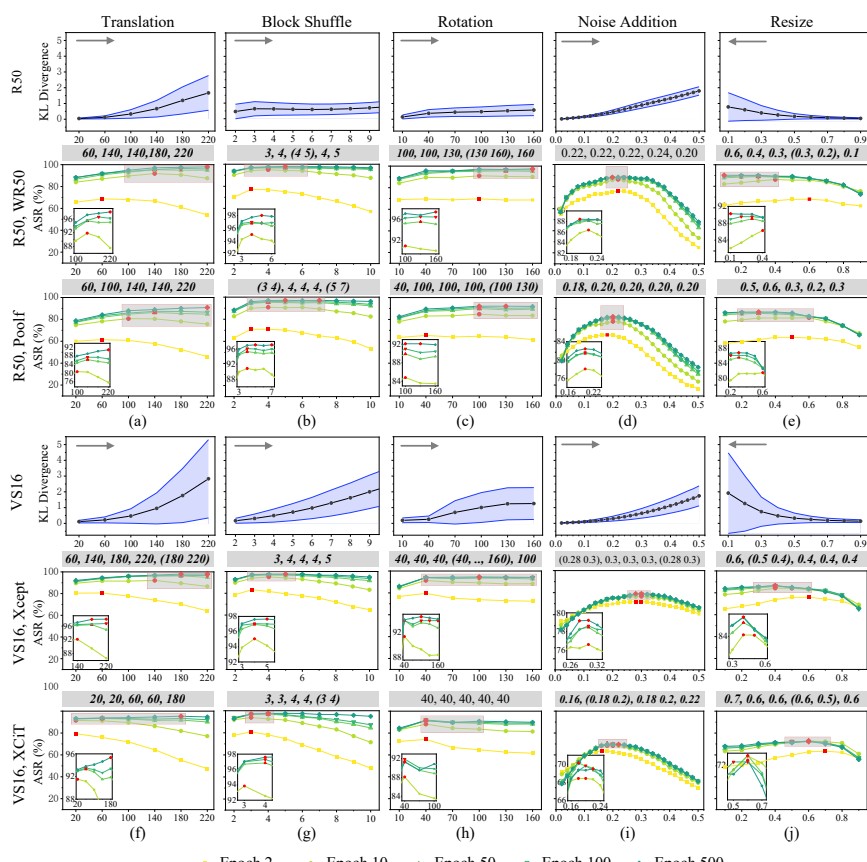

Figure 1: Rows 2, 3, 5, and 6 show the ASRs (%) of various transformation-based attacks integrated with MI-FGSM (left label: model before comma is surrogate, after comma is target). Rows 1 and 4 illustrate the predicted distribution KL divergence of different transformations on benign samples. Red dots indicate the optimal parameters at the corresponding epochs. Values in the gray box indicate the optimal parameters at epochs 2, 10, 50, 100, and 500. Those following the pattern of "optimal transformation parameters grow with iterations" are shown in **_bold italics_**.

## 3.2 EMPIRICAL ANALYSIS

To analyze the dynamics of transferability with varying transformation parameters, we select *Translation*, *Block Shuffle*, *Rotation*, *Noise Addition*, and *Resize* as benchmarks. *Translation* shifts the image horizontally and vertically by $\theta \sim U(0, z)$ (uniform distribution) pixels, where $z$ ranges from 20 to 220 in steps of 40. *Block Shuffle* slices the image with $z$ cuts along each of the horizontal and vertical directions, where $z$ ranges from 2 to 10 in steps of 1. *Rotation* rotates the image by $\theta \sim U(0°, z)$, where $z$ ranges from $10°$ to $160°$ in steps of $30°$. *Noise Addition* adds noise $\theta \sim U(-z, z)$ independently to each element, where $z$ ranges from 0.02 to 0.50 in steps of 0.02. *Resize* resizes the image by a factor $\theta \sim U(z, 1.0)$, where $z$ ranges from 0.1 to 0.9 in steps of 0.1.

Following Equations 3 and 4, assigning the budget $\epsilon = 16/255$, we generate adversarial examples with R50 He et al. (2016) as the surrogate model on the NeurIPS'17 Competition dataset [1] to attack WR50 Zagoruyko & Komodakis (2016) and Poolformer-M36 Yu et al. (2022), and with ViT-S/16 Dosovitskiy et al. (2020) as the surrogate to attack Xception-71 Chollet (2017) and XCiT-S Ali et al. (2021). Experiments are repeated multiple times with random seeds 0 and 1, and the average ASRs are reported in Figure 1 (rows 2, 3, 5, and 6). The following dynamic patterns are observed:

(i) **Transformations leading at low iterations may not retain their advantage at higher iterations.** For illustration, at Epoch 2, with R50 as the surrogate and WR50 as the target, *Noise Addition* (see Fig-

---

[1]https://github.com/anlthms/nips-2017.git

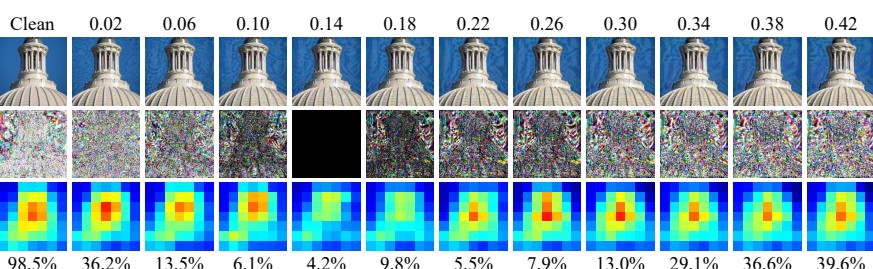

| Clean | 0.02 | 0.06 | 0.10 | 0.14 | 0.18 | 0.22 | 0.26 | 0.30 | 0.34 | 0.38 | 0.42 |

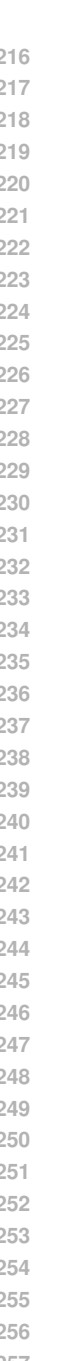

98.5%   36.2%   13.5%   6.1%   4.2%   9.8%   5.5%   7.9%   13.0%   29.1%   36.6%   39.6%

Figure 2: Visualization of the rise-then-fall pattern on individual sample. First row shows *Noise Addition* adversarial examples with R50 as the surrogate at Epoch 500 for different $z$ values (top). Second row shows their absolute differences from the $z = 0.14$ example (normalized by $16/255$). Third row displays deep features from DenseNet161's *denseblock4.denselayer24* (min-max normalization with $\min = -0.0019$ and $\max = 0.0152$). True label probabilities are shown at the bottom.

ure 1(d)) achieves its optimal ASR of 76.0% at $z = 0.22$, while *Translation*, *Rotation*, and *Resize* (see Figure 1(f, h, j)) reach 68.7%, 69.7%, and 68.1%, respectively. However, by Epoch 500, the optimal ASR of *Noise Addition* reaches only 88.5%, whereas those of *Translation*, *Rotation*, and *Resize* are 98.0%, 97.4%, and 90.3%, respectively. Although the other three transformations lag behind *Noise Addition* at low iterations, their ASRs increase rapidly and lead by a large margin at high iterations.

(ii) **Transferability growth over iterations and surrogates differs markedly across parameters. Generally, optimal transformation magnitudes grow with iterations and vary across surrogates.** Taking *Resize* as an example, with ViT-S/16 as the surrogate and XCiT-S as the target (see Figure 1(j)), at $z = 0.9$, the ASRs at Epochs 2 and 500 are 57.0% and 55.9%, respectively, whereas at $z = 0.5$ they are 59.9% and 71.2%. While transferability differences are small at low iterations, larger transformation magnitudes may improve more transferability at higher iterations, whereas inappropriate parameters may cause overfitting and reduce it. In Figure 1, parameters that follow the pattern of optimal transformation magnitudes growing with iterations are highlighted in ***bold italics***. This pattern holds in most cases.

(iii) **Transferability follows a rise-then-fall pattern with respect to transformation magnitude.** This pattern is observed across nearly all transformations and epochs, and is especially clear in the *Noise Addition* (see Figure 1(d, i)). Moreover, this pattern holds not only at the aggregate level but also for individual samples. As shown in Figure 2, for adversarial examples generated by R50 for the same sample under *Noise Addition*-based attacks with varying parameter $z$, deep features of D161 Huang et al. (2017) exhibit a clear rise-then-fall trend, and the predicted probabilities show the same trend.

Although these three patterns appear to describe independent aspects, they are underpinned by a common principle, which motivates a further explanation of transformation-based attacks.

## 3.3 CONCENTRIC DECAY MODEL

A classical explanation for the effectiveness of transformation-based attacks is the Model Augmentation Theory Wang et al. (2021). This theory argues that transformation-based attacks enhance transferability as the combination of transformation and the surrogate forms a new model that emulates a real model. Multiple transformations with varied parameters generate perturbations that transfer across multiple emulated models instead of merely overfitting the surrogate model.

Some of the emulated models are reasonable, in that they are more consistent with the real models trained in practice. We refer to these as *plausible models*. Conversely, the remaining models are classified as *implausible models*. When the combination of transformations and the surrogate emulates plausible models, it facilitates transferability. In contrast, if it emulates implausible models, the resulting perturbations act as noise that does not aid transferability, thereby impairing it. Intuitively, the closer a model's outputs are to those of the surrogate, the more its behavior aligns with reality, and thus it is more likely to be a plausible model. Conversely, a larger discrepancy from the surrogate indicates a higher likelihood of being an implausible model. We are interested in whether the combination of transformations and surrogates is more likely to emulate plausible or implausible models. To this end, we quantify the effect of transformations on the surrogate's outputs with the

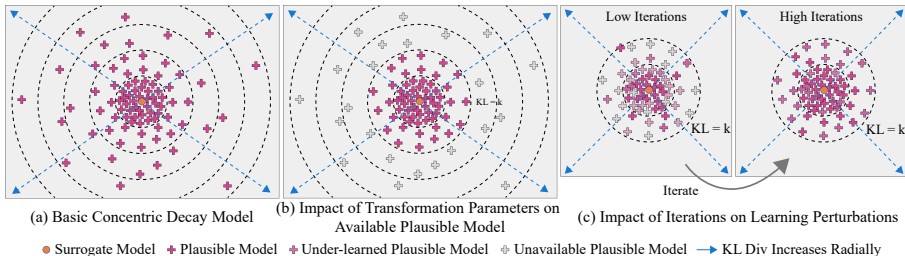

(a) Basic Concentric Decay Model    (b) Impact of Transformation Parameters on Available Plausible Model    (c) Impact of Iterations on Learning Perturbations

● Surrogate Model ✦ Plausible Model ✚ Under-learned Plausible Model ✛ Unavailable Plausible Model → KL Div Increases Radially

Figure 3: Our Concentric Decay Model explains the three revealed patterns. $KL = k$ denotes the high-dimensional surface where the KL divergence with the surrogate model equals $k$.

Kullback–Leibler (KL) divergence, defined as:

$$D_{\mathrm{KL}}[f_S||\mathcal{T}(\boldsymbol{z}) \circ f_S] = \mathbb{E}_{\boldsymbol{x} \sim \mathcal{X}, \boldsymbol{z} \sim \mathcal{Z}}[f_s(\boldsymbol{x}) \log \frac{f_s(\boldsymbol{x})}{\mathcal{T}(\boldsymbol{z}) \circ f_s(\boldsymbol{x})}]. \tag{5}$$

The estimation of KL divergences induced by various transformations are presented in Figure 1 (rows 1, 4), with the number of samples 50. In summary, the density of plausible models around the surrogate decreases concentrically in the KL-divergence space. Smaller transformation magnitudes correspond to smaller KL surfaces, encompassing fewer plausible models. Larger magnitudes yield larger KL surfaces, while covering more plausible models, which also introduce more noise from implausible models, thereby may impair transferability. This is the *Concentric Decay Model*.

As illustrated in Figure 3(a), where the space measured by KL divergence is referenced to the surrogate model, regions closer to the surrogate contain a denser distribution of plausible models, whereas regions farther away are more sparsely populated. Once the transformation parameter $\boldsymbol{z}$ controlling the distribution of the random variable $\boldsymbol{\theta}$ is selected, the maximum KL divergence boundary $k$ between the emulated model and the surrogate model is uniquely determined. As shown in Figure 3(b), the average vulnerability of models within this KL surface can be characterized by adversarial perturbations. With increasing iterations, the adversarial perturbations progressively better approximate the average vulnerability, as illustrated in Figure 3(c).

Therefore, under low iterations, adversarial perturbations insufficiently fit the average vulnerability within surface $KL = k$, preventing them from reflecting the transferability at sufficiently higher iterations, which manifests as Pattern (i). For Pattern (iii), when the transformation magnitude is small, a substantial number of plausible models are still excluded, leaving great room for further transferability improvement. Similarly, when the transformation magnitude is unduly large, an excessive number of implausible models are covered, introducing perturbation noise that attenuates transferability. Hence, an appropriate transformation parameter $\boldsymbol{z}$ is crucial for optimal transferability. Regarding Pattern (ii), for different surrogate models, the distribution of surrounding plausible models varies, so the optimal transformation parameters differ accordingly. Even for the same surrogate model, different numbers of iterations yield different fitting capacities for vulnerabilities. For instance, under low iterations, a large transformation magnitude covers an excessive number of plausible models. Fitting in a dispersed manner dilutes its effectiveness.

### 3.4 DYNAMIC PARAMETER OPTIMIZATION

The three revealed patterns, together with our Concentric Decay Model, suggest the dynamic nature of optimal parameters: optimal parameters vary across surrogate models, iterations, and tasks. So we propose an efficient *Dynamic Parameter Optimization* (DPO) to fully exploit the potential of attacks, which requires adaptive parameter optimization. By contrast, existing attacks configure the same parameters regardless of the surrogate model, iterations, or tasks, which underestimates their full potential. Accordingly, following DPO, we re-optimize existing methods to further validate it.

For baselines, we select attacks spanning different years and representing diverse types, including *Admix* (ICCV'21), *SSIM* (ECCV'22), *STM* (ACMMM'23), and *BSR* (CVPR'24). All other parameters follow the official implementations. The optimized parameters are: admix ratio $\eta$ (0.10 to 0.50, interval 0.02) for Admix, tuning factor $\rho$ (0.3 to 0.9, interval 0.1) for SSIM, noise upper bound $\beta$ (1.2 to 4.0, interval 0.4) for STM, and split number $b$ (1 to 9, interval 1) as well as rotation angle $r$

| | Validation Models | Test Models |
|---|---|---|
| **CNNs** | (1) DenseNet161 | (1) ConvNeXt-B |
| | (2) WR50-2 | (2) VGG-19 |
| | (3) EfficientNet-B0 | (3) IncRes-V2 |
| | (4) Xception-71 | (4) RegNet-X |
| **ViTs** | (5) ViT-B/32 | (5) ViT-B/8 |
| | (6) XCiT-S | (6) SwinT-B/4 |
| | (7) Visformer-S | (7) Convformer-B36 |
| | (8) Poolformer-M36 | (8) Caformer-M36 |

Table 1: Validation models used for optimizing the attacks and test models used to evaluate the optimized results.

---

**Algorithm 1** Refined Dynamic Parameter Optimization

**Require:** dataset D, surrogate model $f_S$, validation models, given configurations $T$, $\epsilon$, $\alpha$ and $N$, attack $A(z_1, ..., z_q)$.
**Ensure:** approximately optimal parameters $z_1^*, ..., z_q^*$.
1: Initiate $z_1^{\text{low}}, ..., z_q^{\text{low}}, z_1^{\text{high}}, ..., z_q^{\text{high}}, z_1 = z_1^{\text{low}}, ..., z_q = z_q^{\text{low}}$.
2: # Repeat the following procedure for each parameter $z_k$
3: **for** $i = 1$ to $\lceil \log_2 m \rceil$ **do**
4:      Generate adversarial example set $D_{\text{adv}}^{\text{low}}$ and $D_{\text{adv}}^{\text{high}}$ by $A(z_1^*, .., z_k^{\text{low}}, .., z_q^{\text{low}})$ and $A(z_1^*, .., z_k^{\text{high}}, .., z_q^{\text{low}})$.
5:      Evaluate $D_{\text{adv}}^{\text{low}}$ and $D_{\text{adv}}^{\text{high}}$ on validation models.
6:      if $z_k^{\text{low}}$ is better than $z_k^{\text{high}}$, $z_k^{\text{high}} = \frac{1}{2}(z_k^{\text{low}} + z_k^{\text{high}})$, v.v.
7: **end for**
8: **return** $z_1^{\text{low}}, ..., z_q^{\text{low}}$.

---

| Attack | Surrogate | Official | $U_2$ | $U_{10}$ | $U_{50}$ | $U_{100}$ | $T_{200}$ | Attack | Surrogate | Official | $U_2$ | $U_{10}$ | $U_{50}$ | $U_{100}$ | $T_{200}$ |
|---|---|---|---|---|---|---|---|---|---|---|---|---|---|---|---|
| Admix $(\eta)$ | R50 | 0.20 | 0.42 | 0.40 | 0.50 | 0.48 | 0.36 | SSIM $(\rho)$ | R50 | 0.5 | 0.7 | 0.7 | 0.8 | 0.8 | 0.5 |
| | ViT-S/16 | 0.20 | 0.30 | 0.38 | 0.42 | 0.40 | 0.42 | | ViT-S/16 | 0.5 | 0.5 | 0.5 | 0.7 | 0.7 | 0.3 |
| | Ens$_{\text{CNNs}}$ | 0.20 | - | - | - | 0.38 | - | | Ens$_{\text{CNNs}}$ | 0.5 | - | - | - | 0.3 | - |
| | Ens$_{\text{ViTs}}$ | 0.20 | - | - | - | 0.50 | - | | Ens$_{\text{ViTs}}$ | 0.5 | - | - | - | 0.5 | - |
| STM $(\beta)$ | R50 | 2.0 | 1.6 | 2.4 | 2.4 | 2.8 | 1.6 | BSR $(b, r)$ | R50 | $2, 24°$ | $3, 20$ | $3, 40$ | $3, 60$ | $4, 60$ | $5, 40$ |
| | ViT-S/16 | 2.0 | 2.8 | 2.8 | 3.2 | 2.8 | 2.4 | | ViT-S/16 | $2, 24°$ | $2, 20$ | $2, 20$ | $2, 40$ | $2, 20$ | $4, 20$ |
| | Ens$_{\text{CNNs}}$ | 2.0 | - | - | - | 1.6 | - | | Ens$_{\text{CNNs}}$ | $2, 24°$ | - | - | - | $2, 20°$ | - |
| | Ens$_{\text{ViTs}}$ | 2.0 | - | - | - | 2.0 | - | | Ens$_{\text{ViTs}}$ | $2, 24°$ | - | - | - | $2, 20°$ | - |

Table 2: Optimized parameters. $U_t$ denotes the optimal parameters of untargeted attacks with $t$ iterations on the validation models. $T_t$ denotes the optimal parameters of targeted attacks with $t$ iterations. - indicates that parameters are not optimized under this condition.

($20°$ to $160°$, interval $20°$) for BSR. Parameters are optimized via a standard grid search: for each surrogate model and iteration number, optimized parameters are varied at fixed intervals, and adversarial examples are evaluated on a validation model set to select the best configuration. Specifically for BSR, we first fix $r = 24°$ and vary $b$. Then, keeping the optimal $b$ fixed, we vary $r$.

We optimize across various tasks. For untargeted single-surrogate attacks, iterations are set to 2, 10, 50, and 100, with surrogate models R50 and ViT-S/16. For untargeted ensemble attacks, optimization is performed at 100 iterations with ensemble surrogate models Ens$_{\text{CNNs}} = \{$R18, R34, R50$\}$ and Ens$_{\text{ViTs}} = \{$ViT-S/16, ViT-S/32, BeiT-B/16$\}$. For targeted single-surrogate attacks, optimization is conducted at 200 iterations for the target label *100*, with surrogate models R50 and ViT-S/16. The validation models employed for parameter optimization are shown in Table 1. The optimized parameters are presented in Table 2. Evaluation in Section 4 demonstrates the success of the optimization.

In practice, as the number of parameters grows, conventional grid search becomes prohibitively costly. For $n$ parameters, each optimized over $m$ steps in $[a, b]$, the precision is $\frac{b-a}{m}$, and the time complexity is $O(mn)$. The rise-then-fall pattern in Pattern (iii) suggests that a bisection-based strategy can accelerate this process, as described in Algorithm 1, reducing the time complexity to $\mathcal{O}(n \log_2 m)$. To account for potential fluctuations, a few additional grid search steps can be applied afterward, achieving precision equal to or better than $\frac{b-a}{m}$. Comprehensive experiments in Section 4 will demonstrate that existing attacks can be optimized with the bisection–based approach.

# 4 EXPERIMENTS AND RESULTS

## 4.1 SETUP

**Configurations** In all experiments, we use a perturbation budget of $\epsilon = 16/255$ with a step size of $\alpha = \epsilon/T$. All experiments are conducted with a random seed of 0. Adversarial examples are generated in parallel on two NVIDIA A100 GPUs and evaluated on a single NVIDIA A100 GPU.

| Attack | Surrogate | Epoch=2 | | Epoch=10 | | Epoch=50 | | Epoch=100 | |
|---|---|---|---|---|---|---|---|---|---|
| | | AVG$_{\text{Validation}}$ | AVG$_{\text{Test}}$ | AVG$_{\text{Validation}}$ | AVG$_{\text{Test}}$ | AVG$_{\text{Validation}}$ | AVG$_{\text{Test}}$ | AVG$_{\text{Validation}}$ | AVG$_{\text{Test}}$ |
| Admix$^\dagger$ | R50 | 56.8$^{\uparrow 2.2}$ | 49.2$^{\uparrow 0.9}$ | 69.8$^{\uparrow 6.2}$ | 61.1$^{\uparrow 5.1}$ | 69.8$^{\uparrow 5.7}$ | 60.5$^{\uparrow 4.7}$ | 68.8$^{\uparrow 4.4}$ | 59.1$^{\uparrow 3.3}$ |
| | ViT-S/16 | 59.9$^{\uparrow 0.4}$ | 54.7$^{\downarrow 0.3}$ | 66.3$^{\uparrow 2.9}$ | 61.0$^{\uparrow 2.0}$ | 65.3$^{\uparrow 3.7}$ | 59.2$^{\uparrow 3.1}$ | 64.7$^{\uparrow 3.7}$ | 58.6$^{\uparrow 3.1}$ |
| SSIM$^\dagger$ | R50 | 66.3$^{\uparrow 0.1}$ | 59.4$^{\downarrow 1.7}$ | 78.6$^{\uparrow 1.1}$ | 72.9 | 84.1$^{\uparrow 2.6}$ | 78.8$^{\uparrow 1.4}$ | 85.8$^{\uparrow 3.6}$ | 80.2$^{\uparrow 2.2}$ |
| | ViT-S/16 | 65.7 | 61.6 | 71.1 | 66.9 | 72.2$^{\uparrow 0.7}$ | 65.8$^{\downarrow 1.6}$ | 72.5$^{\uparrow 0.6}$ | 65.9$^{\downarrow 1.8}$ |
| STM$^\dagger$ | R50 | 70.3$^{\uparrow 0.9}$ | 63.6$^{\uparrow 1.6}$ | 82.3 | 75.0$^{\downarrow 1.7}$ | 87.4$^{\uparrow 0.5}$ | 81.1$^{\downarrow 0.4}$ | 88.0$^{\uparrow 0.5}$ | 80.8$^{\downarrow 1.4}$ |
| | ViT-S/16 | 68.7$^{\uparrow 0.1}$ | 62.5$^{\downarrow 1.7}$ | 79.7$^{\uparrow 0.9}$ | 73.2$^{\uparrow 0.2}$ | 83.0$^{\uparrow 1.8}$ | 76.1$^{\uparrow 0.1}$ | 83.5$^{\uparrow 1.6}$ | 77.4$^{\uparrow 1.3}$ |
| BSR$^\dagger$ | R50 | 70.3$^{\uparrow 1.6}$ | 65.6$^{\uparrow 1.1}$ | 86.8$^{\uparrow 0.9}$ | 83.4$^{\uparrow 1.6}$ | 90.6$^{\uparrow 1.2}$ | 89.2$^{\uparrow 2.4}$ | 91.9$^{\uparrow 1.8}$ | 90.7$^{\uparrow 3.2}$ |
| | ViT-S/16 | 78.6$^{\uparrow 0.8}$ | 73.1$^{\uparrow 1.2}$ | 91.9$^{\uparrow 0.2}$ | 88.1$^{\uparrow 0.3}$ | 94.6$^{\uparrow 0.1}$ | 90.3$^{\downarrow 1.0}$ | 95.0 | 92.0$^{\uparrow 0.2}$ |

Table 3: Average single-surrogate untargeted ASRs (%) of the optimized attacks on the validation and test models. $^\dagger$ indicates results from the optimized methods, $\uparrow$ and $\downarrow$ denote increases and decreases in ASR compared to the unoptimized version, respectively. Improvements are emphasized in bold red. No arrow indicates no change. These notations are consistent across tables.

| Surrogate | Attack | ConvNeXt | VGG19 | IncRes-V2 | RegNet-X | ViT-B/8 | Swin-B | Convformer | Caformer | **AVG** |
|---|---|---|---|---|---|---|---|---|---|---|
| R50 | Admix$^\dagger$ | 9.6$^{\uparrow 2.1}$ | 1.0$^{\uparrow 0.4}$ | 0.5 | 5.1$^{\uparrow 1.9}$ | 0.3$^{\downarrow 0.2}$ | 0.9$^{\uparrow 0.6}$ | 3.1$^{\uparrow 1.0}$ | 1.9$^{\uparrow 0.5}$ | 2.8$^{\uparrow 0.8}$ |
| | SSIM$^\dagger$ | 12.6 | 3.9 | 3.1 | 16.5 | 1.7 | 1.5 | 5.8 | 5.9 | 6.4 |
| | STM$^\dagger$ | 27.9$^{\uparrow 2.8}$ | 5.4 | 7.5$^{\downarrow 1.0}$ | 30.0$^{\uparrow 0.4}$ | 3.9$^{\uparrow 0.4}$ | 3.2$^{\downarrow 0.2}$ | 12.5$^{\downarrow 0.3}$ | 10.5$^{\uparrow 0.5}$ | 12.6$^{\uparrow 0.3}$ |
| | BSR$^\dagger$ | 58.8$^{\uparrow 10.7}$ | 49.8$^{\uparrow 34.1}$ | 11.8$^{\uparrow 6.8}$ | 69.4$^{\uparrow 27.1}$ | 11.4$^{\uparrow 5.7}$ | 9.6$^{\uparrow 4.7}$ | 33.3$^{\uparrow 11.9}$ | 28.7$^{\uparrow 10.2}$ | 34.1$^{\uparrow 13.9}$ |
| ViT-S/16 | Admix$^\dagger$ | 2.4$^{\uparrow 0.9}$ | 0.2$^{\downarrow 0.1}$ | 0.9$^{\uparrow 0.2}$ | 1.9$^{\uparrow 0.7}$ | 3.3$^{\uparrow 0.5}$ | 4.3$^{\uparrow 1.0}$ | 1.5$^{\uparrow 0.6}$ | 3.3$^{\uparrow 0.9}$ | 2.2$^{\uparrow 0.6}$ |
| | SSIM$^\dagger$ | 2.6$^{\downarrow 0.1}$ | 0.5$^{\downarrow 0.3}$ | 0.8$^{\downarrow 0.6}$ | 2.2$^{\downarrow 0.2}$ | 3.2$^{\uparrow 0.9}$ | 4.5$^{\uparrow 0.5}$ | 1.3$^{\uparrow 0.1}$ | 3.9$^{\uparrow 0.9}$ | 2.4$^{\uparrow 0.1}$ |
| | STM$^\dagger$ | 5.6$^{\uparrow 0.1}$ | 0.6$^{\uparrow 0.1}$ | 3.6$^{\uparrow 0.5}$ | 4.6$^{\uparrow 0.8}$ | 6.0$^{\uparrow 0.1}$ | 9.3$^{\downarrow 0.2}$ | 4.1$^{\uparrow 0.6}$ | 9.1$^{\uparrow 1.0}$ | 5.4$^{\uparrow 0.4}$ |
| | BSR$^\dagger$ | 25.3$^{\uparrow 2.1}$ | 10.9$^{\uparrow 4.8}$ | 11.8$^{\uparrow 2.2}$ | 26.4$^{\uparrow 6.5}$ | 22.7$^{\downarrow 0.1}$ | 20.4$^{\uparrow 2.4}$ | 16.3$^{\uparrow 1.2}$ | 25.7$^{\uparrow 0.6}$ | 19.9$^{\uparrow 2.5}$ |

Table 4: Single-surrogate targeted ASRs (%) of the optimized attacks on the test models.

**Dataset and Models.** Following previous work, we conduct experiments on the NeurIPS'17 Competition dataset. The standard test model set without defenses comprises four randomly selected CNN-based models and four attention-based models, as presented in Table 1. Furthermore, we also evaluate the optimized attacks against the defense of adversarial training, with eight state-of-the-art adversarially trained models as baselines: (1) ConvNeXt-L of *MeanS* Amini et al. (2024), (2) ConvNeXtV2-L + Swin-L of *MixedN* Bai et al. (2024), (3) WideResNet-50-2 of *DataF* Chen & Lee (2024), (4) Swin-L of *Charac* Rodríguez-Muñoz et al. (2024), (5) Swin-L of *MIMIR* Xu et al. (2023), (6) ConvNeXt-L of *Compreh* Liu et al. (2025), (7) ConvNeXt-S + ConvStem of *Revisit* Singh et al. (2023), and (8) RaWideResNet-101-2 of *Robust* Peng et al. (2023).

## 4.2 EVALUATION OF OPTIMIZED ATTACKS

### 4.2.1 UNTARGETED SINGLE-SURROGATE ATTACKS.

This experiment aims to investigate whether, when using a single surrogate model, the optimized attacks achieve improved transferability compared to unoptimized attacks in the untargeted setting. The average ASRs on the validation and test model sets are presented in Table 3. Our results indicate that separate optimization across iterations and surrogate models can further enhance transferability, even when the official parameters are near-optimal. Specifically, for BSR at Epoch 100 with R50 as the surrogate model, the unoptimized attack already attained a strong ASR of $87.5\%$, which increased to $90.7\%$ ($+3.2\%$) after optimization. This highlights the importance of dynamic parameter optimization for overcoming transferability limits. Furthermore, the optimal parameters of each attack in Table 2 follow the trend of Pattern (ii), which further validates our theory.

### 4.2.2 TARGETED SINGLE-SURROGATE ATTACKS.

This experiment investigates how dynamic parameter optimization affects the transferability of attacks in the targeted setting. The target label is set to *100*. Table 2 indicates that the optimal parameters differ notably between untargeted and targeted attacks. For example, with R50 as the surrogate, targeted attacks require much smaller parameters than untargeted attacks for Admix, SSIM, and STM, while BSR exhibits the reverse. Table 4 reports the complete results on the test models. Re-

| Attack | Ens$_{\text{CNNs}}$ | | Ens$_{\text{ViTs}}$ | | Attack | Ens$_{\text{CNNs}}$ | | Ens$_{\text{ViTs}}$ | |
|---|---|---|---|---|---|---|---|---|---|
| | AVG$_{\text{Validation}}$ | AVG$_{\text{TEST}}$ | AVG$_{\text{Validation}}$ | AVG$_{\text{TEST}}$ | | AVG$_{\text{Validation}}$ | AVG$_{\text{TEST}}$ | AVG$_{\text{Validation}}$ | AVG$_{\text{TEST}}$ |
| Admix† | 89.8$^{\uparrow 1.3}$ | 82.2$^{\uparrow 1.8}$ | 88.5$^{\uparrow 3.4}$ | 83.3$^{\uparrow 3.0}$ | SSIM† | 93.0$^{\uparrow 0.2}$ | 87.9$^{\uparrow 0.2}$ | 92.4 | 88.2 |
| STM† | 94.3$^{\uparrow 0.3}$ | 88.4$^{\uparrow 0.8}$ | 94.5 | 91.0 | BSR† | 95.8 | 92.7$^{\uparrow 0.2}$ | 98.5$^{\uparrow 0.1}$ | 96.4$^{\uparrow 0.3}$ |

Table 5: Average ASRs(%) on validation and test models of ensemble-surrogate untargeted attacks.

| Epoch | Surrogate | Attack | MeanS | MixedN | DataF | Charac | MIMIR | Compreh | Revisite | Robust | **AVG** |
|---|---|---|---|---|---|---|---|---|---|---|---|
| 10 | R50 | Admix† | 12.5$^{\uparrow 1.4}$ | 12.0$^{\uparrow 0.7}$ | 24.4$^{\uparrow 1.4}$ | 10.8$^{\uparrow 0.6}$ | 10.2$^{\uparrow 0.6}$ | 12.3$^{\uparrow 1.2}$ | 17.4$^{\uparrow 1.0}$ | 18.5$^{\uparrow 0.3}$ | 14.8$^{\uparrow \mathbf{0.9}}$ |
| | | SSIM† | 15.9$^{\downarrow 0.2}$ | 15.5$^{\uparrow 0.3}$ | 27.5$^{\uparrow 0.4}$ | 16.0$^{\uparrow 0.7}$ | 14.0$^{\uparrow 0.4}$ | 15.5$^{\downarrow 0.4}$ | 22.8$^{\uparrow 0.4}$ | 21.2$^{\uparrow 0.3}$ | 18.6$^{\uparrow \mathbf{0.3}}$ |
| | | STM† | 18.8$^{\uparrow 1.4}$ | 17.4$^{\uparrow 0.5}$ | 31.3$^{\uparrow 1.3}$ | 17.8$^{\uparrow 0.9}$ | 15.5$^{\uparrow 0.6}$ | 18.4$^{\uparrow 1.4}$ | 24.2$^{\uparrow 1.1}$ | 24.1$^{\uparrow 1.2}$ | 20.9$^{\uparrow \mathbf{1.0}}$ |
| | | BSR† | 14.7$^{\uparrow 0.3}$ | 14.3 | 26.8$^{\uparrow 0.7}$ | 13.9$^{\downarrow 0.3}$ | 12.1$^{\uparrow 0.1}$ | 14.0$^{\uparrow 0.1}$ | 20.6$^{\uparrow 1.3}$ | 20.9$^{\uparrow 0.8}$ | 17.2$^{\uparrow \mathbf{0.4}}$ |
| | ViT-S/16 | Admix† | 13.3$^{\uparrow 0.6}$ | 13.7$^{\uparrow 0.4}$ | 25.7$^{\uparrow 0.8}$ | 13.8 | 12.4$^{\downarrow 0.2}$ | 13.1$^{\uparrow 0.4}$ | 21.1$^{\downarrow 0.1}$ | 20.2 | 16.7$^{\uparrow \mathbf{0.3}}$ |
| | | SSIM† | 14.9 | 15.5 | 26.8 | 15.9 | 14.0 | 13.9 | 21.9 | 20.9 | 18.0 |
| | | STM† | 17.4$^{\uparrow 1.6}$ | 18.6$^{\uparrow 1.6}$ | 29.7$^{\uparrow 2.0}$ | 20.3$^{\uparrow 3.1}$ | 17.6$^{\uparrow 2.2}$ | 17.1$^{\uparrow 1.8}$ | 25.4$^{\uparrow 2.0}$ | 23.8$^{\uparrow 2.2}$ | 21.2$^{\uparrow \mathbf{2.0}}$ |
| | | BSR† | 16.1$^{\downarrow 0.1}$ | 18.2$^{\uparrow 0.7}$ | 29.3$^{\downarrow 0.1}$ | 19.2$^{\uparrow 0.3}$ | 14.9$^{\uparrow 0.2}$ | 16.0$^{\uparrow 0.4}$ | 23.3 | 21.6$^{\downarrow 0.7}$ | 19.8 |
| 100 | R50 | Admix† | 11.8$^{\uparrow 1.1}$ | 11.5$^{\uparrow 0.5}$ | 24.0$^{\uparrow 1.0}$ | 10.5$^{\uparrow 0.8}$ | 9.9$^{\uparrow 0.9}$ | 11.5$^{\uparrow 1.5}$ | 16.3$^{\uparrow 0.5}$ | 18.6$^{\uparrow 1.4}$ | 14.3$^{\uparrow \mathbf{1.0}}$ |
| | | SSIM† | 17.4$^{\uparrow 0.7}$ | 16.6$^{\uparrow 0.4}$ | 29.0$^{\uparrow 1.3}$ | 17.5$^{\uparrow 0.2}$ | 14.9$^{\uparrow 0.8}$ | 16.9$^{\uparrow 0.5}$ | 23.6$^{\uparrow 1.1}$ | 23.0$^{\uparrow 1.2}$ | 19.9$^{\uparrow \mathbf{0.8}}$ |
| | | STM† | 21.7$^{\uparrow 2.4}$ | 19.8$^{\uparrow 1.0}$ | 35.7$^{\uparrow 3.2}$ | 22.7$^{\uparrow 2.2}$ | 18.5$^{\uparrow 1.7}$ | 20.9$^{\uparrow 1.9}$ | 26.8$^{\uparrow 2.3}$ | 26.9$^{\uparrow 2.5}$ | 24.1$^{\uparrow \mathbf{2.1}}$ |
| | | BSR† | 15.4$^{\uparrow 0.2}$ | 15.4$^{\uparrow 0.9}$ | 28.5$^{\uparrow 2.0}$ | 15.9$^{\uparrow 0.5}$ | 13.5$^{\uparrow 0.9}$ | 15.2 | 21.4$^{\uparrow 1.9}$ | 22.0$^{\uparrow 0.6}$ | 18.4$^{\uparrow \mathbf{0.9}}$ |
| | ViT-S/16 | Admix† | 12.6$^{\uparrow 0.7}$ | 13.1$^{\uparrow 0.5}$ | 26.0$^{\uparrow 1.2}$ | 12.6$^{\downarrow 0.6}$ | 11.9$^{\downarrow 0.2}$ | 12.2$^{\uparrow 0.1}$ | 20.8$^{\uparrow 1.2}$ | 19.5$^{\uparrow 0.6}$ | 16.1$^{\uparrow \mathbf{0.4}}$ |
| | | SSIM† | 14.7$^{\uparrow 0.5}$ | 15.7$^{\uparrow 0.4}$ | 27.3$^{\uparrow 0.7}$ | 15.6 | 13.8$^{\uparrow 0.5}$ | 14.2$^{\uparrow 0.4}$ | 22.7$^{\uparrow 1.1}$ | 21.3$^{\uparrow 1.1}$ | 18.2$^{\uparrow \mathbf{0.6}}$ |
| | | STM† | 17.8$^{\uparrow 1.9}$ | 19.1$^{\uparrow 2.0}$ | 30.2$^{\uparrow 1.8}$ | 20.8$^{\uparrow 3.0}$ | 17.7$^{\uparrow 2.4}$ | 17.4$^{\uparrow 2.4}$ | 25.8$^{\uparrow 2.2}$ | 24.1$^{\uparrow 2.1}$ | 21.6$^{\uparrow \mathbf{2.2}}$ |
| | | BSR† | 16.9$^{\uparrow 0.1}$ | 17.9$^{\uparrow 0.1}$ | 29.7 | 20.5$^{\uparrow 0.8}$ | 15.7$^{\downarrow 0.1}$ | 16.7 | 25.2$^{\uparrow 0.5}$ | 22.7$^{\uparrow 0.1}$ | 20.7$^{\uparrow \mathbf{0.1}}$ |

Table 6: Untargeted ASRs (%) on attacking advanced adversarial trained models.

markably, with R50 as the surrogate model, the optimized BSR attack achieves an average ASR gain of 13.9% on test models, including a dramatic 34.1% increase against VGG-19. This underscores the importance of dynamic parameter optimization for advancing transferability and points to the strong potential of transformation-based attacks in targeted scenarios.

### 4.2.3 UNTARGETED ENSEMBLE-SURROGATE ATTACKS.

This experiment investigates the difference when transformation-based attacks are combined with other strategies (e.g., ensemble-based strategies). We adopt Ens$_{\text{CNNs}}$ and Ens$_{\text{ViTs}}$ as ensemble surrogates. The optimized parameters, presented in Table 2, differ from the official ones in most cases. The average ASRs on validation and test models are reported in Table 5, showing that optimized attacks achieve either improvements or no loss compared to the official configurations.

### 4.2.4 ATTACKS AGAINST ADVERSARIAL TRAINING DEFENSES.

The above experiments are conducted on standard models. To further evaluate the performance of attacks against defended models, we consider eight advanced adversarially trained models as targets and re-run untargeted attacks using the parameters optimized at $T = 100$. Table 6 shows that optimization substantially increases the average ASR across the eight advanced adversarially trained models, demonstrating the value of dynamic parameter optimization against defended models.

## 5 CONCLUSIONS

This work presents the first study of the dynamics of transferability with respect to parameters, revealing three patterns. The Concentric Decay Model (CDM) is proposed to bridge the gap in existing research. The revealed patterns and our CDM indicate the dynamic nature of optimal parameters. Motivated by this, we propose an efficient Dynamic Parameter Optimization (DPO) to dynamically optimize existing attacks. Extensive experiments demonstrate that the re-optimization significantly improves transferability. By leveraging the rise-then-fall pattern and a bisection-based approach, our DPO reduces the complexity from $\mathcal{O}(mn)$ to $\mathcal{O}(n \log_2 m)$. However, our investigation is capped at 500 iterations. The dynamics at higher iterations remain an intriguing topic for future study.

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

## A  THE USE OF LARGE LANGUAGE MODELS

In this work, we only employ large language models (ChatGPT-5) for basic polishing of human-written text (e.g., minor adjustments of wording or sentence order to improve readability) and for grammar checking. No ideas, analyses, or experiments are generated or conducted by LLMs. All contributions are proposed and validated by the authors.

## B  SUPPLEMENTARY MATERIALS

### B.1  DETAILED RESULTS OF UNTARGETED ATTACKS WITH VARYING PARAMETERS

Section 4.2.1 reports the average untargeted single-surrogate ASR of re-optimized attacks. Here, we provide detailed results under varying parameters to show the landscape. Figure 4 shows the average non-targeted ASRs of adversarial examples generated by *Admix*, *SSIM*, *STM*, and *BSR* on a single surrogate model with varying parameters, evaluated across eight validation models in Table 1.

These results suggest that the dynamics of transferability in existing transformation-based attacks are highly consistent with the three patterns revealed in this work. Notably, despite minor fluctuations, different attacks generally follow the rise-then-fall pattern across iterations and surrogate models, validating the feasibility of DPO with a bisection-based strategy.

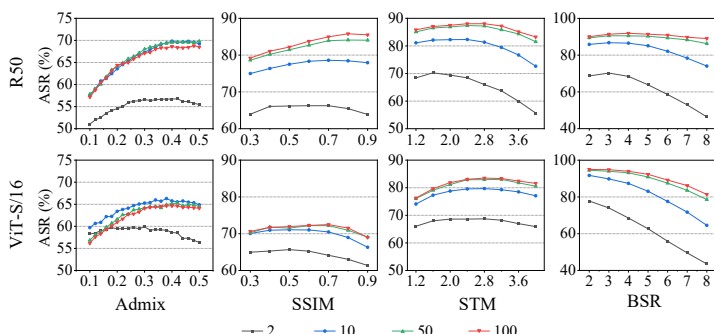

Figure 4: Average untargeted ASRs (%) of existing attacks with R50 or ViT-S/16 as the surrogate model under varying parameters against eight validation models.

## B.2 VISUALIZATION OF ADVERSARIAL EXAMPLES GENERATED ON OPTIMIZED ATTACKS

In this part, we provide the illustration of adversarial examples of untargeted single-surrogate attacks generated with both the official and the re-optimized parameters, as shown in Figure 5 and Figure 6.

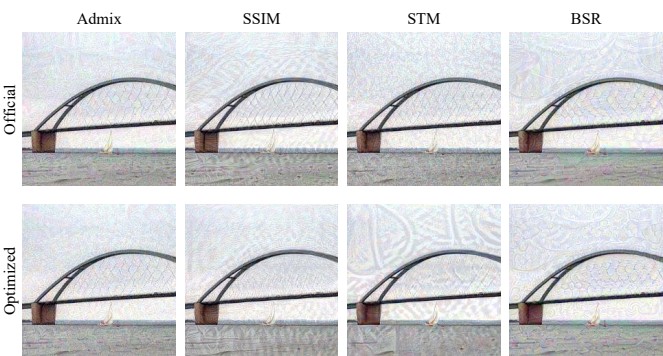

Figure 5: Visualization of adversarial examples generated on untargeted attacks with R50 as the surrogate model. Official indicates the examples generated on official parameters. Optimized indicates the examples generated on re-optimized parameters. The number of iterations is 100.

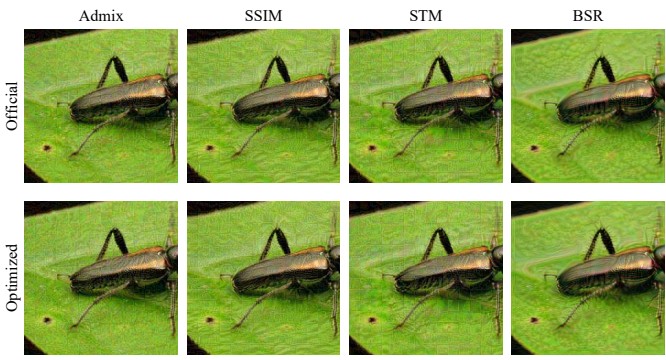

Figure 6: Visualization of adversarial examples generated on untargeted attacks with ViT-S/16 as the surrogate model. The number of iterations is 100.

