# OpenReview forum: "Dynamic Parameter Optimization for Highly Transferable Transformation-based Attacks"
_ICLR.cc/2026/Conference — ICLR 2026 Conference Withdrawn Submission_

### Official Review · Reviewer_jY71 · 2025-10-26

**Soundness:** 2
**Presentation:** 3
**Contribution:** 2
**Rating:** 4
**Confidence:** 5

**Summary:**

This paper systematically investigates the dynamic impact of parameters on the transferability of transformation-based attacks, revealing that the optimal parameters vary across different surrogate models, target models, and iteration counts. To address this, the authors propose an efficient Dynamic Parameter Optimization (DPO) method, which also reduces the computational complexity of parameter search.

**Strengths:**

1、The paper is well-structured, clearly written, and easy to follow.

2、The proposed Concentric Decay Model (CDM) intuitively quantifies the relationship between transformation strength, transferability, and the number of iterations.

**Weaknesses:**

1、The method requires the use of validation models for parameter selection. Since these validation models are already utilized, why not integrate them into the surrogate model ensemble to further enhance the transferability of adversarial examples? This could potentially lead to higher transferability.

2、The time complexity should be compared with existing transfer-based attack methods, not just grid search. Existing methods do not require searching over iterations and transformation parameters, while the proposed method introduces significant additional computational overhead. Is this trade-off justified? In what real-world scenarios would this be applicable?

3、According to Table 3, improvements are mostly around 1%, which is relatively modest, though in some cases reach over 5%, showing notable potential.

**Questions:**

1、Clearly state the specific improvement percentages in the abstract (e.g., "significantly improves by X%").

2、Provide a detailed description of Algorithm 1 in the main text.

---

### Official Review · Reviewer_bPK9 · 2025-10-31

**Soundness:** 3
**Presentation:** 2
**Contribution:** 2
**Rating:** 4
**Confidence:** 3

**Summary:**

This paper investigates the transferability of transformation-based adversarial attacks against deep neural networks. The authors empirically observes several patterns in the dynamics of transformation-based attacks, notably that number of iterations has different impacts on different attacks, transformation magnitudes tend to grow with number of iterations, and transferability of attacks tends to follow an inverted U pattern with number of iterations. The concentric decay model is then presented to explain these patterns. Experiments are run on a variety of neural network architectures showing that optimization can enhance attack success rate for transformation based attacks.

**Strengths:**

* This work shows how the strength of certain types of adversarial attacks is currently underestimated and could have implications for how defenses to adversarial examples are evaluated.
* Many different types of transformation-based attacks are used in evaluation. These attacks have also been well-studied in prior work in the machine learning robustness literature, and it is clear how this paper relates to prior research in the field.
* The concentric decay model seems plausible and gives a convincing intuition for why additional iterations might improve the performance of these attacks.
* The evaluations are thorough, investigating many attack types, transferability settings, and DNN architectures.

**Weaknesses:**

* The evaluations don't directly test the validity of the concentric decay model. The model would be more convincing if there were estimates of the divergences that are discussed in Section 3.3.
* The time complexity of this approach is not thoroughly discussed. While it is shown that the DPO approach improves over the naive approach to parameter optimization in terms of time complexity, there are no theoretical or experimental results showing how much time this approach adds over unoptimized transformation-based attacks. Given the high number of iterations that are discussed, it may be the case that this makes adversarial example generation significantly slower.
* Included results are only based off individual runs, and no estimates of error are reported. Some of the reported improvements are close to zero and could reasonably be the result of noise.
* The text doesn't emphasize enough the heterogeneity in the experimental results. There are many instances in which optimized attacks perform worse than the unoptimized attacks (although those instances are in the minority). Also, averaging attack performance over different models doesn't seem entirely appropriate in this setting. I would appreciate if there were some intuition provided for when unoptimized attacks outperform their optimized counterparts, as that is an unintuitive phenomenon.

**Questions:**

* What is dynamic about dynamic parameter optimization? Is it just the fact that parameters are optimized after each iteration?
* In algorithm 1, how are $z^{low}$ and $z^{high}$ set?
* Why does the rise-then-fall pattern suggest the use of the binary search in Algorithm 1? The rise-then-fall pattern is with respect to the number of iterations, while Algorithm 1 is optimizing attack parameters.
* In Algorithm 1, how many adversarial examples must be generated on each iteration in order to separate $x_k^{low}$ and $z_k^{high}$ with statistical significance?
* Did you test whether these techniques can be used to improve adversarial training?

---

### Official Review · Reviewer_PeZV · 2025-11-02

**Soundness:** 3
**Presentation:** 3
**Contribution:** 2
**Rating:** 2
**Confidence:** 4

**Summary:**

This paper examines transformation-based adversarial attacks and points out that existing methods use poorly optimized parameters, especially under different iteration settings and models. The authors analyze how transferability changes with transformation strength, explain it using a Concentric Decay Model, and propose a Dynamic Parameter Optimization method to improve efficiency and performance. Experiments show clear gains in transferability across models and tasks.

**Strengths:**

This paper is grounded in empirical observations, analyzing the limitations of adversarial perturbations under three dynamic patterns and introducing a clear perspective to relate surrogate models with emulated plausible or implausible models. Based on this analysis, the work proposes a well-motivated Dynamic Parameter Optimization method that efficiently improves cross-model transferability, providing both conceptual insights and practical contributions.

**Weaknesses:**

1. The CDM model is based on the similarity between surrogate models and emulated models, measured via KL divergence, but several concerns arise:
- The increase in KL divergence does not appear to significantly reflect the coverage of plausible models for the three dynamic patterns; the results in Figure 1 do not convincingly support this analysis. Large KL differences do not clearly introduce excessive noise that reduces transferability.
- It is unclear whether there is any theoretical analysis or formal mathematical proof supporting this CDM model.
- The role and significance of the parameter K in KL divergence are not addressed in the experiments.
2. Can the transferability between different model architectures(CNNs to ViTs or ViTs to CNNs) be compared to further validate the generalization of the proposed method?
3. The current two-step parameter search in BSR is relatively simple. Are there more efficient optimization methods that could improve the search process?
4. It would be useful to provide PSNR, SSIM, or other image quality metrics to better assess attack quality and image fidelity.

Typos:
+ The references at line 71 and line 72 seem to be wrong.

**Questions:**

Please refer to weakness

---

### Official Review · Reviewer_tfFm · 2025-11-04

**Soundness:** 3
**Presentation:** 3
**Contribution:** 2
**Rating:** 2
**Confidence:** 4

**Summary:**

This paper explores the dynamic parameter optimization for transformation-based transferable adversarial attacks. The perspective is interesting, as previous studies typically regard these parameters as hyperparameters, which are fixed for different surrogate models and iterations. However, the observation in this paper (for example, the results in Figure 1) didn't motivate me. Moreover, the improvement of the attack success rate is somewhat low.

**Strengths:**

1. This is an interesting research, targeting the parameter optimization of transformation-based adversarial attacks.
2. The authors provided some empirical results to explain their motivation, making the writing clear for readers.

**Weaknesses:**

1. The observation in this paper isn't very motivated for me. In Figure 1, the selection of epoch 2 is too extreme, since the adversarial examples may not converge in a few iterations. For epochs 10, 50, 100, and 500, although the optimal parameters differ, the ASRs remain relatively stable within the gray box (i.e., across the different optimal parameters). This makes me doubt the importance of this paper.
2. The experimental results are somewhat low, with most improvements of the AVG being lower than 1%. The limited improvement also validates the stable attack performance within the gray box shown in Figure 1.

**Questions:**

I agree with the opinion that the optimal parameters are different for different surrogate models; however, when attacking different test models, do the optimal parameters also differ?

---

### Official Review · Reviewer_2zyG · 2025-11-12

**Soundness:** 2
**Presentation:** 2
**Contribution:** 2
**Rating:** 4
**Confidence:** 4

**Summary:**

This paper focuses on a transformation-based approach, and challenge the use of inappropriate hyperparameter setting. From the perspective of model augmentation and the KL divergence, the authors propose the Concentric Decay Model (CDM) to effectively explain these patterns. They also propose an efficient Dynamic Parameter Optimization (DPO) to reduce the complexity

**Strengths:**

Adversarial transferability is an important topic.

Using the model augmentation to explain is interesting.

**Weaknesses:**

Need a discussion on the balance between performance and time complexity.

The connection between the KL divergence and the plausible models should be further explained.

The experiment only focuses on the transformation-based approach independently.

**Questions:**

1 This paper focuses on the setting of large iterations. Why is a large iteration important? Additionally, as shown in Figure 1, the improvement of increasing iterations is limited (50=>100, even sometimes hurts the performance). Considering the performance and the time complexity of a large iteration, the authors should  further discuss this issue.

2 Similarly, we can observe a slight performance gain in the experiment. The authors should discuss the time complexity of the proposed DPO quantitatively, e.g., report the time cost in the experiment and the comparisons with baselines.

3 The connection between the KL divergence and the plausible models is not quite clear. Why are they correlated, can you give any explanations or proofs?

4 As shown in Figure 1, the patterns of different transformation-based approaches are not quite the same. How do you solve this problem when we consider using multiple transformation-based approaches together?

---

### Note · Authors · 2025-11-25

I have read and agree with the venue's withdrawal policy on behalf of myself and my co-authors.